Multi-step partitioning combined with SOM neural network-based clustering technique effectively improves SAT solver performance

Yun Siyu
Wang Xinsheng xswang@hit.edu.cn
School of Information Science and Engineering, Harbin Institute of Technology at Weihai , Weihai, Shandong , China
Angiulli Giovanni
Electronic publication date: 2025 Aug 14
Publication date: 2025
Volume: 11
Electronic Location ID: e3076
Received 2025 Feb 7; Accepted 2025 Jul 3
Copyright: © 2025 Yun and Wang
Copyright year: 2025
Copyright holder: Yun and Wang
License: This is an open access article distributed under the terms of the Creative Commons Attribution License, which permits unrestricted use, distribution, reproduction and adaptation in any medium and for any purpose provided that it is properly attributed. For attribution, the original author(s), title, publication source (PeerJ Computer Science) and either DOI or URL of the article must be cited.
License URL: https://creativecommons.org/licenses/by/4.0/

Keywords: SAT, SOM neural network, Cluster, Divide, Structural information

Funding: The authors received no funding for this work.

==============================
As the core engine of electronic design automation (EDA) tools, the efficiency of Boolean Satisfiability Problem (SAT) solver largely determines the cycle of integrated circuit research and development. The effectiveness of SAT solvers has steadily turned into the key bottleneck of circuit design cycle due to the dramatically increased integrated circuit scale. The primary issue of SAT solver now is the divergence between SAT used in industry and research on pure solution algorithms. We propose a strategy for partitioning the SAT problem based on the structural information then solving it. By effectively extracting the structure information from the original SAT problem, the self-organizing map (SOM) neural network deployed in the division section can speed up the sub-thread solver’s processing while avoiding cumbersome parameter adjustments. The experimental results demonstrate the stability and scalability of our technique, which can drastically shorten the time required to solve industrial benchmarks from various sources.

Introduction

The first nondeterministic polynomial time (NP)-complete problems to be established is the Boolean satisfiability problem (SAT) problem (Cook, 1971). Nowadays, systematic search algorithms and stochastic search algorithms are the two main types of solutions for SAT problems. The former typically combines Davis–Putnam–Logemann–Loveland (DPLL) (Davis & Putnam, 1960), conflict driven clause learning (CDCL) (Claessen et al., 2008), Look Ahead (LA) (Heule & Van Maaren, 2006) and other strategies to conduct a depth-first search on the entire independent variable assignment space. The latter usually randomly generates assignment of all variables, flips some variables and tests the satisfiability, then carries out the above process until the upper limit of iteration times is reached. Stochastic search algorithms usually combined with strategies such as survey propagation (SP) (Braunstein, Mézard & Zecchina, 2005), message passing (MP) (Zhao et al., 2011), evolutionary algorithm (EA) (Wang, Fan & Li, 2014) and so on. The combination algorithm, which combines systematic search and stochastic search, has demonstrated great solution efficiency recently (Cai & Zhang, 2021).

SAT problems are currently used in a variety of practical contexts, including logic synthesis, Boolean matching (Safarpour et al., 2006), circuit equivalence checking (Goldberg, Prasad & Brayton, 2001), and model checking. The discipline of EDA greatly benefits from the effective resolution of SAT problems. The solution time under the assumption of specific computing resources is the primary optimization content in the present research on SAT solving methods. An efficient SAT solving algorithm will significantly shorten software and hardware design time.

In the history of computers science, the divide-and-conquer algorithm has been a popular technique for handling complicated issues. Divide-and-conquer algorithms are used in both Quick Sort algorithms and fast Fourier transform. In the field of very-large-scale integration (VLSI) design, it is usually necessary to convert large-scale integrated circuits into small scales for layout planning, wiring, etc. (Alpert, Mehta & Sapatnekar, 2008). In the field of SAT, the original SAT instance was split up into multiple smaller SAT problems by Fan et al. (2014). Unfortunately, the similarity threshold and other parameters must be set repeatedly for SAT instances in different application scenarios, therefore the algorithm application’s generalizability is low. The SAT problem is created as a hypergraph in the article (Duraira & Kalla, 2004), and the original problem is divided using the hypergraph partition method. Although changing the order of independent variable assignment in accordance with the partition result can significantly speed up solutions, the algorithm’s effectiveness is constrained by how well the hypergraph partitioning tool works. By giving a variable a value, Nair et al. (2022), Le Frioux et al. (2019), and Hyvärinen, Junttila & Niemelä (2011) divide the search area into several sections. In order to expedite the solution, methods like sharing learned clauses are introduced while the two parts are being solved in parallel. The strategy of reducing time by increasing memory occupation is too intensive for computer hardware. In the encoding process from constraint satisfaction problem (CSP) problem to SAT problem, Velev & Gao (2009) used a hierarchical method to partition the domains of CSP variables, obtaining efficient conjunctive normal form (CNF) encoding results and significantly improving the speed of solving the original problem. But this method only applies to CSP problems and does not involve solving more general SAT problems.

We enhanced the SAT problem partitioning technique and included the clause learning mechanism in the solution process based on the solution framework suggested in article (Fan et al., 2014). Following high-dimensional mapping of the SAT problem represented by the original CNF format, we clustered the multi-clusters after clustering using the SOM network, then divided the original SAT problem into two clusters using the simulated annealing approach. After the sub-thread SAT solver call and the inter-cluster clause learning process, finally the solution results are obtained.

Experiments show that our algorithm can solve SAT problems in industrial instances more efficiently, and the solution time is shorter than calling other solvers directly. In addition, our algorithm is faster on average than the article (Fan et al., 2014). Furthermore, This algorithm can be effectively integrated with any other SAT solver because it doesn’t alter the sub-thread solver.

The remainder of this article is structured as follows: The fundamentals of solving the SAT instances after partitioning and the basics of using a SOM neural network for clustering are introduced in the “Preliminaries” “Proposed Approach” is the proposed approach. The effectiveness of using our approach and other techniques is contrasted in “Experimental Results” “Conclusion” concludes the article and outlines future work.

Preliminaries

SAT solution based on clause cluster

The study of Fan et al. (2014) proposes a divide and conquer strategy for tackling SAT problems, which is the foundation for all of our research. Industrial practical problems are usually composed of multiple modules, such as a central processing unit (CPU) circuit consisting of modules such as controllers, arithmetic logic units, registers, etc. The logic gates within the modules are closely connected, while the two logic gates in different modules have relatively small connections. The existence of modules generates structural information for actual problems. Similarly, SAT problems transformed from industrial practical problems will also contain this structural information. However, the traditional SAT algorithm directly solves the actual problem after converting it into the CNF format, which leads to the structure information not being used. Theoretically and experimentally, it has been demonstrated that partitioning the original SAT instance into many clusters and solving each cluster individually may considerably increase the algorithm’s efficiency if the partition is reasonable (Fan et al., 2014). The theoretical analysis is as follows:

For example, a SAT instance in the form of CNF is shown in Eq. (1).

(1) F=(x1∨x2∨x3)∧(¬x1∨x2∨¬x3∨x4)∧(x2∨x3∨¬x4)∧(¬x3∨x4∨x5).

It contains five variables and four clauses. If the original instance is divided into the following two clusters: (x1∨x2∨x3), (¬x1∨x2∨¬x3∨x4) and (x2∨x3∨¬x4), (¬x3∨x4∨x5), Considering that the variables shared by these two clusters are x2 and x3, if the two variables x2 and x3 are first assigned, the two divided clusters can be solved separately. If both clusters are satisfiable, then the entire problem is satisfiable. If one of the cluster is unsatisfiable, then re-assign x2 and x3 and repeat the above-mentioned solving process separately. If all the assignments up to x2 and x3 have been traversed but there is no satisfiable clusters, then the original SAT problem is unsatisfiable.

The theoretical solution time of direct solving and solve after division are as follows: Assuming that the original SAT problem contains Nsum variables, the first cluster contains N1 variables, the second cluster contains N2 variables, and the common variables of the two clusters are Ncom, obviously Eq. (2) holds.

(2) Nsum=N1+N2−Ncom.

Since each variable has only two assignments of true or false, after using the divide and conquer method, the solution time of the original SAT problem is expressed by Eq. (3).

(3) Tdiv=ts×2Ncom×((2N1−Ncom)+(2N2−Ncom))=ts×(2N1+2N2).

The direct solution time is given by Eq. (4).

(4) Tdir=ts×2Nsum

where ts is the time for a single assignment. Since both N1 and N2 are smaller than Nsum in most cases, Tdiv is usually less than Tdir, i.e., as long as the division is reasonable, the total solution time will be shortened.

From the analysis above, it can be seen that the division method of the original SAT instance is one of the most important elements in this procedure, we propose a method to divide the original SAT problems efficiently, which shortens the solution time dramatically.

SOM neural network

The self-organizing map (SOM) neural network was described as a competitive learning unsupervised neural network by Kohonen (1990). It has two layers, an input layer and an output layer, both of which are completely interconnected. The output layer neurons have a certain topology, with planar structures being the most prevalent, while the number of neurons in the input layer is equal to the size of the input vector. A vector with the same length as the number of input layer neurons makes up each output layer neuron’s feature. In the scenario of clustering, neurons in the input layer are set as the dimension of the samples to be clustered, and neurons in the output layer are set as the number of target clusters when utilized for clustering. The samples are fed into the network, which is then trained. The clustering procedure comes to a finish when the output layer neurons after training correlate to various samples. The SOM network exhibits great generalization capabilities throughout the clustering process and is particularly well suited for extracting topological structure from high-dimensional samples. Figure 1 depicts the SOM network structure with a plane output layer, where xi(i=1,2,…,n) represents each dimension of the n-dimensional vector to be clustered.

Figure 1 SOM network structure diagram.

The network clustering process is as follows: 1. Initialize the number of neurons in the input layer, the topology and number of neurons in the output layer, and the feature vector of each neuron in the output layer.

2. Input all the samples to be clustered according to a certain order, find the output layer neuron closest to each sample as the winning neuron, and update the feature vectors of this neuron and some surrounding neurons.

3. After the training process, input all samples into the network again, and record the neuron closest to each sample. For the same neuron in output layer, all recorded samples are the content of this cluster.

In the following section, we will show how to use SOM network to cluster the clauses in a STA problem and the effectiveness of this approach.

Proposed approach

The overview of our approach is shown in Fig. 2. Our algorithm takes the SAT file as input and output the solving result of this problem finally. Firstly we conduct preliminary cluster on the original problem to aggregate the clauses with subset relationship. Secondly, we use SOM neural network to cluster the clauses further. Minimum cut division is used to form two sets of clauses thirdly, which results in two smaller SAT files. In the solving procedure, a sub-thread solver is adopted to solve two smaller SAT problem during which learned clauses are shared between two SAT problems. If the original SAT problem is satisfiable, our algorithm will give a satisfying assignment.

Figure 2 The overview of our partition and solving procedure.

Each green oval denotes a clause in original SAT problem.

Multi-step partition

Division process

Multi-level division techniques have been widely applied in the field of circuit division, particularly VLSI division (Cong & Lim, 2004; Saab, 2004; Li & Behjat, 2006). The integrated circuit is first transformed into a hypergraph, where nodes represent circuit modules and hyperedges represent signal nets in the circuit. Initially, two or more highly correlated vertices are coalesced, and this contraction process is iteratively repeated until the original hypergraph is mapped to a significantly smaller hypergraph. Following an initial partitioning of this reduced hypergraph, the contracted nodes are progressively uncoarsened. During this uncoarsening phase, the initial partition is refined, ultimately achieving the partitioning of the entire large-scale hypergraph. We apply this multi-step division method to the SAT problem. Here is the specific division procedure: 1. Preliminary clustering. Each clause of the SAT instance in CNF format is taken as an element for preliminary clustering. That is to say, write the set of all clauses containing independent variables respectively, and aggregate any two clauses with subset relations into one category. For example, a clause 1: (x1∨x2) contains the set of independent variables {x1,x2}, clause 2: (x2∨¬x3) contains the set of independent variables {x2,x3}, clause 3: (x1∨x2∨¬x3) contains the set of independent variables {x1,x2,x3}, since {x1,x2} is a subset of {x1,x2,x3}, and {x2,x3} is a subset of {x1,x2,x3}, then these three clauses are clustered into one category.

2. SOM clustering. Take each cluster after preliminary clustering as a sample and input it into the SOM network for clustering. Use each cluster of the SOM network clustering result as a new sample, and apply the SOM network again for clustering. Details of using SOM neural network are in “Clustering With SOM Network”.

3. Minimum cut division. When the total number of clusters of the SOM network clustering is smaller than 100, the clustering is stopped, and the simulated annealing algorithm (Steinbrunn, Moerkotte & Kemper, 1997) is used for minimum cut division. Firstly, the output layer of the SOM network is divided into two parts based on the diagonal, named CA and CB. Each cluster is assigned to either CA or CB according to its position, as shown in Fig. 3. Then randomly move a cluster from CA to CB, calculate whether the number of the cut-set variables between CA and CB is reduced, and if so, keep this operation, if the number of cut-set variables does not decrease, keep this operation with a certain probability. Repeat the cluster assignment between CA and CB until the number of iterations reaches the limit or the annealing process ends.

Figure 3 Schematic diagram of creating CA and CB after SOM clustering, where Ci(i=1…17) represents each cluster of SOM clustering results.

Finally, the original SAT problem is divided into two large clusters, CA and CB, and the division step ends. The flowchart of multi-step division is as Fig. 4.

Figure 4 Flowchart of multi-step division.

Min-cut and bisection

In the third step in division process, simulated annealing is used for minimum cut division instead of bisection1 . The specific reasons are as follows:

Assuming that the original SAT problem is divided into CA with N1 variables, CB with N2 variables, and cut-set variables between the two clusters with Ncom. Consider the most extreme min-cut division situation: that is, CB contains only one clause, And CA contains all the remaining clauses, and the total time for solving at this time will not be greater than directly calling the sub-thread solver to solve. If the method of bisection is used, the total theoretical solution time is proportional to 2N1+2N2−Ncom×p, where p represents the number of cut-set variable assignments that may make CA satisfiable. Since there are Ncom cut-set variables, p is related to 2Ncom, and its specific value is associated with the specific SAT problem and the division method, so it is difficult to measure accurately. As a result, the application of bisection will produce unstable efficiency and significant fluctuations in the total solution time. We employ the minimum cut approach to guarantee that the algorithm has a somewhat stable efficiency improvement even in the worst-case scenario.

Advantages of modular solution

Because SOM network has the ability of clustering based on structural information, each step of multi-step clustering can extract the structural information of the original SAT problem at a certain level, and finally divide the original SAT into multiple clusters. Due to the correspondence between clauses in the CNF format SAT problem and modules in industrial practical problems, this step also divides the practical problem into multiple modules. Our approach divides the original problem into two parts finally, which are solved separately to reduce the scale of the original problem and improve the efficiency of solution.

Figure 5 is a schematic diagram of the modular division of the practical problem, where “Module3” is the module that makes the corresponding SAT problem unsatisfiable. In this case, solving CB will directly lead to unsatisfiable, thus reducing the solution time.

Figure 5 Schematic diagram of multi-step division of the original SAT problem.

Rectangles of different sizes represent modules of different levels, which can be obtained by different steps of SOM cluster.

Clustering with SOM network

In this subsection, We will discuss how to use SOM neural network to cluster clauses and why SOM neural network can leverage structure information from industrial problem effectively.

Space mapping ideas

According to the usage guidelines of SOM neural networks, each clause must be mapped to a point in a high-dimensional space in order to cluster all the clauses. These are the precise steps: 1. The number of variables in this SAT problem is used as the dimension of the target space, that is, the variables correspond to the dimensions one by one.

2. If a clause contains variable xi, then the value of the point corresponding to this clause is 1 on this dimension, otherwise it is 0.

Examples are as follows:

For SAT instance Eq. (5):

(5) F=(x1∨x2∨¬x3)∧(x1∨¬x2)∧(¬x2∨x3)∧(x1∨x3)

Eq. (5) contains three variables and four clauses (C1: (x1∨x2∨¬x3), C2: (x1∨¬x2), C3: (¬x2∨x3), C4: (x1∨x3)), so the target space is three-dimensional, the number of samples is 4. The four points of the mapping result are: (1, 1, 1), (1, 1, 0), (0, 1, 1), (1, 0, 1). Its position in three-dimensional space is shown in Fig. 6, where X1, X2 and X3 represent three dimensions respectively.

Figure 6 The position of clauses after spatial mapping.

Since the distance between (1, 1, 0) and (1, 1, 1) is 1, and the distance between (1, 1, 0) and (0, 1, 1) is 2, obviously, the latter distance is longer. This is also consistent with the intuitive feeling of the similarity between clauses (there are two common variable between C1 and C2, and there is one common variables between C2 and C3). After mapping all the clauses of a SAT instance, they can be input into the SOM network for clustering.

Parameter selection

The output layer’s topology is set to square to reduce the amount of calculation. An empirical formula states that it has a strong ability to cluster when the number of output layers is 5n (Shalaginov & Franke, 2015), where n is the number of samples. To simplify the calculation, we adopt triangle function to update the feature of neurons in output layer.

Periodic boundary modification

After analyzing the mechanism of using SOM network to conduct clustering tasks, we note there are some drawbacks in traditional output layer, to enhance the ability of clustering, We modify the topological relationship of neurons in the boundary layer of the SOM network, and apply it to the subsequent clustering process. There are two reasons that motivate us to make this modification.

Reason 1: Throughout the training phase, the characteristics of the neurons in the network output layer will continue to converge on the average value of one or more samples. As a result of this procedure, the neuron has a higher likelihood of being the final topological clustering location for these samples. Following training, the neurons in the output layer typically exhibit higher resemblance to the peripheral neurons (Kohonen, 1990). It is possible to think of numerous neurons with similar spacing as forming a whole. It is important to note that the neurons in the center of the entire partially convey its properties. If, during training, a neuron that belongs in the center is moved to the output layer’s edge, other neurons that belong nearby will find it challenging to connect with it, which will result in the loss of some structural information.

Reason 2: Compared to the intermediate neurons in the output layer, the neighbors of the neurons near the edge of the output layer are fewer (there are three or five neurons in a circle around the neurons at the edge, and 8 neurons in a circle around the neurons at the inside). which means that throughout the entire network training period, they will receive less updates than the internal neurons. The training’s overall clustering effect will be somewhat impacted by this mismatch in the amount of updates. As a result, edge neurons’ neighbor properties must be modified.

To make the neurons at boundary have the same property as internal neurons, We logically connect the leftmost neurons of the square output layer with the rightmost neurons, and logically connect the upmost neurons with the downmost neurons. As shown in Fig. 7, after modification, neurons A and B become neighbors.

Figure 7 Neighborhood diagram.

Before periodic boundary modification, neuron (A) and (B) are not neighbor while they become neighbor after modification.

The frequency of each neuron’s victories both before and after the periodic border change is shown in Fig. 8. It is clear that the modified output layer no longer has a logical boundary and that, during training, it shares the same status as an internal neuron.

Figure 8 Win map comparison.

The darker the color, the more times the neuron wins. The color corresponds to the number of wins of the neuron.

Solving process after partition

Instead of comparing the solving results of sub-problems directly by Fan et al. (2014), which may waste much time on explored solution space. We introduce a clause learning mechanism in the process of solving the SAT problem after division to form a systematically search algorithms.

Firstly, the sub-thread solver is used to solve the SAT problem corresponding to CA. If the solution result is UNSAT, the original SAT problem is UNSAT. If the solution result of CA is SAT, then all the cut-set variable assignments are substituted into CB, and then the sub-thread solver is used to solve the problem CB. If the CB cluster is SAT, the original SAT problem is satisfiable. If the CB cluster cannot be satisfied, it means that the assignment of the cut-set variable at this time will conflict. The above process needs to be re-iterated. In order to prevent the CA cluster from outputting the previous cut-set variable assignment results again, the clauses learned in the previous iteration need to be added to the CA. For example, when CA is SAT, the cut-set variables are x1, x2, and x3, and the output result is x1=1; x2=0; x3=1, but this assignment results in CB being unsatisfied, then add a clause (¬x1∨x2∨¬x3) to CA. Secondly, exchange the positions of CA and CB, repeat the above process, until a certain iteration, both CA and CB are SAT or the first cluster is UNSAT, the solution ends.

The flow chart of solving two clusters is shown in Fig. 9, where CA/CB refers to CA or CB, and the positions are exchanged after each clause learning process.

Figure 9 Flow chart of solution process.

Experimental results

Clustering results

In terms of obtaining structural information, the SOM network outperforms the strategy suggested in the article (Fan et al., 2014) by a wide margin. In contrast to the article (Fan et al., 2014), which necessitates numerous adjustments to various parameters for different benchmarks, including similarity threshold and minimum cluster number. In addition, clustering SAT problems with huge scale or close numbers of variables in each clause may result in bad effect. However, SOM neural networks can overcome this flaw, as demonstrated by the following example:

In the equivalence test of combinational logic circuits, a common method is to connect the circuit to be optimized and the optimized circuit into a miter structure (Brand, 1993). That is to say, join the same input ports of two circuits together, connect the identical output ports to an XOR gate, connect the outputs of all of these newly connected XOR gates to an OR gate, and utilize the output of the OR gate as the miter circuit’s overall output. The circuits before and after optimization are not equivalent if there is an input assignment that causes the final output to be true. If a SAT solver is used to solve this inspection procedure, each gate (AND, OR, NOT, and XOR) must be translated into a clause. These clauses are then joined by conjunctions, and the output port of the miter circuit is set to true. The two circuits are equivalent if the SAT problem is unsatisfiable; otherwise, they are not equivalent.

Figure 10 depicts a four-bit array multiplier with eight input ports and eight output ports, which is made up of 16 one-bit multipliers. As several of the original circuit’s pins had a 0 as their input, we simplify it to a circuit made up of nine one-bit multipliers and three full adders. After using the method in article (Fan et al., 2014) to convert the circuit to a SAT problem, it has a total of 253 clauses and 85 variables.

Figure 10 The structure diagram and cell circuit of a four-bit array multiplier.

Using the SOM network and the method described in the article (Fan et al., 2014), clustering was carried out. The results demonstrate that the SOM network produces 40 clusters, each of which has been colored to indicate which module it belongs to (one-bit multiplier or full adder), as shown in Fig. 11. This means that the SOM neural network clusters logic gate which belong to the same module to nearby neurons, further, it can leverage the structure information from real-world SAT problem.

Figure 11 Corresponding position of circuit module in output layer.

It can be seen that neurons belonging to the same module are all adjacent to each other.

However, when the clustering threshold is adjusted to 0.51, the approach described in Fan et al. (2014) produces a clustering result of 62 clusters. The clustering outcome is 1 cluster when the threshold is set to 0.5. The modular information of the original circuit is challenging to replicate. It is evident that the SOM network is proficient at extracting structural information.

Multi-step division results

To form a fair comparison with Fan et al. (2014), we use the the same benchmarks, namely SATLIB (Hoos & Stützle, 2000) and SAT competition 2013 Balint et al. (2013a, 2013b) to conduct experiments, and list some of the division results in Table 1, where the first column is the file name, we abbreviated part of the file name2 , the second column is the number of variables, and the third column is the number of clauses, and the fourth column is the number of cut-set variables of the CA and CB after partition. It can be seen from Table 1 that in top ten benchmark, which are from industrial problem, the number of cut-set variable are less than 5% of variables in original problem. While in the last two benchmarks, which are generated randomly, the number of cut-set variables takes more than 5% of original problem variables. This is because the random-generated problems don’t have structure information thus they can’t be divided by modules.

Table 1 Division results of some benchmarks.

File name	Variables	Clauses	Cut-set variables	
bivium-40-200-0s0-0xd4	972	5,450	13	
bivium-40-200-0s0-0x92	970	5,432	11	
bivium-40-200-0s0-0x66	971	5,469	10	
bivium-39-200-0s0-0xdc	974	5,505	13	
bivium-39-200-0s0-0x5f	978	5,514	13	
bivium-39-200-0s0-0x53	971	5,434	17	
bivium-39-200-0s0-0x28	973	5,507	9	
bivium-39-200-0s0-0x1b	972	5,432	12	
am_6_6.shuffled-as.sat03-362	2,269	7,814	12	
aes_24_4_keyfind_2	520	5,968	16	
unif-k3-r4.25-v360-c1530-S14488	360	1,530	23	
unif-k5-r21.3-v90-c1917-S11595	90	1,917	28	

To demonstrate the role of each part in the multi-step partition procedure, Table 2 shows the number of total clusters, max cluster size and cut-set variables changes in one representative benchmark3 . As the process of division moves forward, it can be seen that the total number of clusters declines while the number of variables in the largest cluster rises, which means our multi-step partition method cluster the original problem gradually and each step contributes to the final division result.

Table 2 Number of clusters and cut-set variables after each step.

Step	Total clusters	Max cluster size	Cut-set variables	
Original file	5,450	5	591	
Preliminary cluster	886	5	591	
First time SOM	62	745	457	
Second time SOM	30	745	450	
Minimum cut divide	2	832	12	

Time comparison

To compare the time used by our algorithm and other method, we conduct experiments on three benchmarks, namely SATLIB (Hoos & Stützle, 2000), SAT competition 2013 Balint et al. (2013a, 2013b), and SAT competition 2023 (Balyo et al., 2023). The first two benchmarks are selected because they are also used in Fan et al. (2014), the third benchmarks are the latest open source benchmark from SAT competition. In terms of sub-thread solver, we select MiniSat (Giunchiglia & Tacchella, 2004), Glucose (Audemard & Simon, 2018) and CaDiCaL (Biere et al., 2024). MiniSat is one of the most well-known commercial solvers, which performs exceptionally well in industrial scenarios. Glucose is a derivative of MiniSat, who won 8 years of SAT competition since emergence. CaDiCaL is one of the most modern SAT solvers and its variant won first place in the 2023 SAT Competition in main track. The maximum number of iterations for the simulated annealing algorithm is set to 5,000 in “Min-cut and Bisection”. All experiments are conducted on a Intel Core i7 11800H machine with 16 GB of memory and Ubuntu 18.04 operating system.

There are more than one thousand benchmarks in the three benchmarks suite, running all the benchmarks will be a time consuming task, so we select all the benchmarks with less than 10,000 clauses to conduct comparison experiment. Our study focus on the effectiveness of partition before solving instead of mechanism of different sub-thread solver, so we compare the performance between partition before solving and directly solving. Table 3 shows the result of four comparisons across four benchmark suites. Winning rate denotes the portion of benchmarks our algorithm use less time than directly solving, speed up rate denote the average speed up rate on wining benchmarks. ‘SOM’ denotes using SOM neural network to conduct partition while ‘Sim’ denotes using partition methods from Fan et al. (2014). From Table 3 we can see that in three benchmark suites except ‘SC2013-Random’, our method perform better than directly solving to a large extent when using different sub-thread solvers, which means our partition algorithm is suitable for various solvers. In ‘SC2013-Random’ benchmark suite, we only speed up on a small percentage of benchmarks, this is because the random-generated SAT problems lack of structure information form real-world problem, thus a reasonable division of these problems is hard to achieve, making it difficult to reduce the overall solution time. In addition, our partition method outperforms that of Fan et al. (2014) dramatically; our method speed up more than 69% instances of three industrial benchmark suites while that of Fan et al. (2014) speeds up less than 30%. Which demonstrates the practicality of our algorithm.

Table 3 Winning rate and speed up rate of different methods across benchmarks.

Methods	SC2013-Industry	SC2013-Random	SATLIB	SC2023	
SOM+MiniSAT WR(%)	73.2	12.4	69.5	80.6	
SOM+MiniSAT ASU(%)	59.3	21.3	53.7	42.6	
Sim+MiniSAT WR(%)	21.4	5.3	15.2	26.4	
Sim+MiniSAT ASU(%)	18.5	8.0	24.3	13.7	
SOM+Glucose WR(%)	76.8	8.9	72.8	69.2	
SOM+Glucose ASU(%)	48.2	15.5	42.0	55.1	
SOM+CaDiCaL WR(%)	68.3	16.0	37.8	77.3	
SOM+CadiCaL ASU(%)	53.7	5.3	129.3	64.9	
Note:

WR denotes wining rate, ASU denotes average speed up rate, SC denotes SAT Competition.

Table 4 shows detail results of some benchmarks where ‘SOM+solver’ represents dividing first and then calling the corresponding solver to solve, the time unit is seconds. From the table, it can be seen that our algorithm significantly improves the solving efficiency for most industrial examples. The method of partitioning first and solving later did not show advantages for instances ‘am_5_5.shuffled-as.sat03-361’, ‘vmpc_29.cnf’ and ‘571a2…9e’, which may be due to the special structure of the instances. At the same time, it can be seen that there is not much difference between the pure solving time (total solution time minus partition time) of these two instances and directly calling the solver to solve, which is due to the use of the min-cut mechanism in the “Min-cut and Bisection”. This proves that using the min-cut instead of bisection, our algorithm will not significantly prolong the solution time in extreme situations. The last two rows in Table 4 show the results of random-generated instance, our method exhibits inferior performance relative to direct solving, as it prioritizes structural information extraction from stochastic problem instances, which maybe meaningless.

Table 4 Detail solving time results between different methods.

File name	Source	Partition time	MiniSat	SOM+MiniSAT	Glucose	SOM+Glucose	CaDiCaL	SOM+CaDiCaL	
bivium-40-200-0s0-0xd4	Industry	31.70	1,805.62	1,207.89	419.41	133.74	634.24	452.63	
bivium-40-200-0s0-0x92	Industry	31.87	2,827.12	1,770.66	663.17	254.65	424.46	254.73	
bivium-40-200-0s0-0x66	Industry	32.77	738.06	517.91	738.06	563.24	534.63	453.04	
am_6_6.shuffled-as.sat03-362	Industry	362.51	2,416.29	1,895.61	231.96	489.35	842.32	534.25	
am_5_5.shuffled-as.sat03-361	Industry	38.43	1.01	39.90	1.48	39.47	1.21	42.01	
aes_16_10_keyfind_3	Industry	623.90	1,425.45	844.25	189.48	705.34	254.73	1,045.32	
aes_24_4_keyfind_2	Industry	155.47	1,371.62	237.68	2,535.31	2,368.56	437.34	205.24	
aes_24_4_keyfind_4	Industry	154.71	129.03	168.46	308.36	233.74	245.89	182.45	
aes_32_3_keyfind_1.cnf	Industry	26.77	470.04	573.78	318.99	152.32	402.53	32.44	
aes_32_3_keyfind_2.cnf	Industry	26.57	308.74	139.88	162.46	145.79	294.63	127.43	
aes_64_1_keyfind_1.cnf	Industry	16.52	47.73	45.68	>3,000	122.50	142.74	114.68	
vmpc_29.cnf	Industry	505.60	141.29	539.42	24.59	535.64	253.64	104.63	
vmpc_30.cnf	Industry	639.41	2,439.61	1,355.21	115.43	743.21	353.08	823.52	
6s165.cnf	Industry	325.50	673.41	449.46	28.61	362.48	3.23	337.46	
644fd…58	Industry	8.27	835.53	352.43	1,103.53	639.23	235.72	143.89	
af1e8…c5	Industry	28.36	2,053.52	1,245.26	843.12	254.23	515.42	354.39	
571a2…9e	Industry	13.74	10.32	23.59	3.62	16.34	1.53	15.79	
73063…64	Industry	16.50	3,942.64	154.34	823.46	643.57	158.73	89.49	
dd169…05	Industry	6.44	672.53	475.27	1,473.46	642.67	734.62	41.53	
d17aa…f0	Industry	37.05	1,235.43	637.85	384.24	135.32	303.35	243.77	
unif-k3-r4.25-v360-c1530-S14488	Random	28.52	143.11	540.76	480.68	2,462.83	1,457.35	1,643.64	
unif-k5-r21.3-v90-c1917-S11595	Random	21.11	679.02	271.07	2,543.58	2,804.95	154.32	257.45	
Note:

The best results are shown in bold.

Table 5 shows detail results of different partition methods. The first column in Table 5 is the file name, the second column is the source of each benchmark, the third column is time used for the whole partition, the fourth column is the solution time of directly calling MiniSat, the fifth column is the total solution time using our partition algorithm, and the sixth column shows the solution time using the partitioning algorithm in article (Fan et al., 2014). Table 5 shows that compared to the algorithm in article (Fan et al., 2014), our algorithm exhibits greater acceleration in most industrial instances except ‘am 6 6.shuffledas.sat03362’ and ‘571a2…9e’. For instance ‘am 6 6.shuffledas.sat03362’, this is mainly due to the large number of variables and clauses (2,269 variables and 7,814 clauses) it contains, making partitioning time a bottleneck in its efficiency. While for ‘571a2…9e’, the directly solving time is less than our partition time, which showcases the limitation of our algorithm on special problems. In addition, Similar to our algorithm, Fan et al. (2014) also perform poorly on random benchmarks which means the idea of extracting structure information from STA problem only works on industrial problems.

Table 5 Solving time comparison between different partition methods.

File name	Source	Partition time	MiniSat	SOM+MiniSat	Sim+MiniSat (Fan et al., 2014)	
bivium-40-200-0s0-0xd4	Industry	31.70	1,805.62	1,207.89	1,634.53	
bivium-40-200-0s0-0x92	Industry	31.87	2,827.12	1,770.66	2,389.40	
bivium-40-200-0s0-0x66	Industry	32.77	738.06	517.91	547.49	
bivium-39-200-0s0-0xdc	Industry	31.74	867.004	752.39	834.52	
644fd…58	Industry	8.27	835.53	352.43	753.97	
af1e8…c5	Industry	28.36	2,053.52	1,245.26	2,586.36	
571a2…9e	Industry	13.74	10.32	23.59	8.64	
73063…64	Industry	16.50	3,942.64	154.34	2,563.44	
am_6_6.shuffled-as.sat03-362	Industry	362.51	2,416.29	1,895.62	1,598.43	
aes_24_4_keyfind_2	Industry	155.47	1,371.62	237.60	409.64	
unif-k3-r4.25-v360-c1530-S14488	Random	28.52	143.11	540.76	463.51	
unif-k5-r21.3-v90-c1917-S11595	Random	21.11	679.02	271.07	356.42	
Note:

The best results are shown in bold.

Conclusion

Researchers have become interested in SAT because of its unique position in the field of propositional reasoning and search. Currently, research on effective SAT problem-solving algorithms will significantly shorten the hardware design cycle due to the growing size of integrated circuits and the dramatic rise in testing and verification time. It has major implications for the information industry.

We employ the SOM neural network in the clustering procedure for the division of the SAT problem with the aim of effectively exploiting the structure information in real-world industrial applications. We effectively extract the structure information while avoiding the challenging tuning work. The partition of the SAT instance is approached using the multi-step partition concept from the VLSI domain. The original SAT problem is split into two clusters using a combination of preliminary clustering and multi-step SOM clustering. Then, the simulated annealing algorithm is used to find the minimum cut. Finally, the clause learning mechanism is added during the process of calling the sub-thread solver to solve each cluster separately. The results of the final experiment demonstrate that this method can significantly increase the efficiency of the SAT solution.

Our technique reduces the time spent on parameter adjusting, but it also lengthens the division time. As a result, for large-scale problems, the division time may be more than the direct solution time, resulting in a decrease in efficiency. In future work, we will explore a faster clustering algorithm while maintaining its accuracy. The clustering based on structural information process is the most important aspect of this technique and efficiency of the total solution will be directly impacted by the clustering outcome. For problems in different fields, the final level of abstraction required is different. As a result, when to stop SOM clustering and begin min-cut partition needs further research.

Supplemental Information

Supplemental Information 1 The entire program used to run on Windows.

Supplemental Information 2 The entire program used to run on Linux.

Supplemental Information 3 Benchmark for SAT solving experiments with 596 variables and 2,780 clauses.

Supplemental Information 4 Benchmark for SAT solving experiments with 520 variables and 2,968 clauses.

Supplemental Information 5 Benchmark for SAT solving experiments with 2,269 variables and 7,814 clauses.

Supplemental Information 6 Benchmark for SAT solving experiments with 972 variables and 5,432 clauses.

Supplemental Information 7 Benchmark for SAT solving experiments with 978 variables and 5,514 clauses.

Supplemental Information 8 Benchmark for SAT solving experiments with 973 variables and 5,570 clauses.

Supplemental Information 9 Benchmark for SAT solving experiments with 971 variables and 5,434 clauses.

Supplemental Information 10 Benchmark for SAT solving experiments with 974 variables and 5,505 clauses.

Supplemental Information 11 Benchmark for SAT solving experiments with 971 variables and 5,469 clauses.

Supplemental Information 12 Benchmark for SAT solving experiments with 970 variables and 5,432 clauses.

Supplemental Information 13 Benchmark for SAT solving experiments with 972 variables and 5,450 clauses.

Supplemental Information 14 Benchmark for SAT solving experiments with 286 variables and 1,742 clauses.

Supplemental Information 15 Benchmark for SAT solving experiments with 2810 variables and 11,683 clauses.

Supplemental Information 16 Benchmark for SAT solving experiments with 360 variables and 1,530 clauses, seed = 144,043,535.

Supplemental Information 17 Benchmark for SAT solving experiments with 360 variables and 1,530 clauses, seed = 1,448,866,403.

Supplemental Information 18 Benchmark for SAT solving experiments with 90 variables and 1,917 clauses.

Additional Information and Declarations

Competing Interests

The authors declare that they have no competing interests.

Author Contributions

Siyu Yun conceived and designed the experiments, performed the experiments, analyzed the data, performed the computation work, prepared figures and/or tables, and approved the final draft.

Xinsheng Wang conceived and designed the experiments, authored or reviewed drafts of the article, and approved the final draft.

Data Availability

The following information was supplied regarding data availability:

The code is available in the Supplemental Files.

1 The meaning of minimum cut here is to minimize the number of variables in the partition result, while the meaning of bisection is to make the number of variables in the two clusters approximately equal.

2 For example, unif-k5-r21.3-v90-c1917-S1159550149-077.cnf is abbreviated as unif-k5-r21.3-v90-c1917-S11595.

3 The file name is bivium-40-200-0s0-0xd447c33176b6b675fd5f8dc3a5deda46569dc34eedf37da020-6.cnf.

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
