# Peer review of "Multi-step partitioning combined with SOM neural network-based clustering technique effectively improves SAT solver performance"

_PeerJ Computer Science, doi:10.7717/peerj-cs.3076_

## Round 0.1 · original submission · Major Revisions

Dear Authors,
Your paper has been revised. It needs major revisions before being accepted for publication in PEERJ Computer Science. More precisely:

1) You must improve the overall presentation of your approach. There's some repetition when describing the model and its components, which must be avoided. Your ideas must be presented more clearly and concisely to make them more understandable for other researchers and practitioners.

2) it would be helpful to see how your method performs on different SAT problems.

3) The claim that "the theoretical calculation time of this solution process is less than that of the original SAT problem" must be clarified. Furthermore, it would be informative to evaluate or comment on the approach's performance on all SATLIB and SC benchmarks rather than a small selected subset.

·

Basic reporting

The paper describes a novel strategy for increasing SAT solver performance using multi-step partitioning and self-organizing maps (SOM). The writing is generally clear, using professional, straightforward English. The introduction clearly establishes the problem that the study seeks to address, highlighting the difference between SAT applications in industry and academic research. Relevant literature is cited, including early research on the SAT problem as well as current advances that are consistent with disciplinary norms.

Experimental design

The research question is clear and focuses on improving SAT solver techniques for increasingly complex integrated circuits. However, the way the proposed method is presented could be more organized and easier to follow. There’s some repetition when describing the model and its components, which could be avoided. While the use of a Self-Organizing Map and a clause learning mechanism is interesting, presenting these ideas more clearly and concisely would help highlight the strengths of the approach and make it more understandable for other researchers and practitioners.

Validity of the findings

The results are backed by strong experiments, showing that the proposed method improves the SAT solver, with faster solving times and better clustering. The findings are reliable enough to be tested and repeated by other researchers. However, it would be helpful to see how the method performs on different types of SAT problems. While the authors explain why the results matter, an additional investigation of how the strategy might operate in diverse settings over time would make the conclusions even stronger.

Additional comments

Overall, the paper presents a compelling advancement in SAT-solving strategies. The authors effectively demonstrate the benefits of their multi-step partitioning approach through a variety of benchmarks. However, it would be beneficial to include discussions of limitations and scenarios where the proposed method may struggle, particularly with randomly generated SAT problems, as noted in the findings. Furthermore, suggestions for future research on optimizing the clustering process or exploring alternative neural network architectures could provide valuable direction.

Reviewer 2 ·

Basic reporting

The paper proposes a SAT-solving workflow that first clusters SAT clauses using a high-dimensional SOM, applies a multi-step partitioning process, and then solves the subproblems separately while sharing learned clauses. The solving procedure is based on a divide-and-conquer method proposed by Fan et al. (2014), which is different from the mainstream CDCL-based approach. The method is similar in spirit to theory combination, where a problem is partitioned into clusters. Each cluster is tackled individually, until a satisfying assignment that all clusters agree on is found or all possible arrangements of shared variable assignments are exhausted.

A key challenge in this process is the initial problem partitioning, and the paper presented a new SOM-based partitioning approach that is different from prior similarity-based partitioning approaches and appears to perform better on the tested set of benchmarks.

Experimental design

The paper evaluates their approach on selected SATLIB and SAT Competition 2013 benchmarks. And compare their approach with a number of baselines: most directly, Fan et al. with a prior partitioning strategy, as well as other SAT solvers such as Lingeling and Glucose. Both the benchmarks and the solvers are from more than a decade ago. It would be helpful to evaluate all benchmarks from those benchmark sets rather than a selected subset.

Validity of the findings

As far as I can tell, the experimental evaluation validated the claim that the new SOM-based approach can speed up the divide-and-conquer-based solving proposed in Fan et al (2014) at least in some cases.

Additional comments

- The claim that "the theoretical calculation time of this solution process is less than that of the original SAT problem" needs to be clarified.
- It would be informative to evaluate or comment on the performance of the approach on all SATLIB and SC benchmarks, rather than a small selected subset.

---

## Round 0.2 · accepted · Accept

Dear Authors,
Your paper can be accepted for publication in PEERJ Computer Science. Thank you for your fine contribution.